# Stimulus-induced gamma rhythms are weaker in human elderly with mild cognitive impairment and Alzheimer's disease

Dinavahi VPS Murty[1], Keerthana Manikandan[1], Wupadrasta Santosh Kumar[1], Ranjini Garani Ramesh[1], Simran Purokayastha[1], Bhargavi Nagendra[1], Abhishek ML[1], Aditi Balakrishnan[1], Mahendra Javali[2], Naren Prahalada Rao[3], Supratim Ray[1]*

[1]Centre for Neuroscience, Indian Institute of Science, Bengaluru, India; [2]MS Ramaiah Medical College & Memorial Hospital, Bengaluru, India; [3]National Institute of Mental Health and Neurosciences, Bengaluru, India

*For correspondence:
sray@iisc.ac.in

**Competing interests:** The authors declare that no competing interests exist.

## Abstract

Alzheimer's disease (AD) in elderly adds substantially to socioeconomic burden necessitating early diagnosis. While recent studies in rodent models of AD have suggested diagnostic and therapeutic value for gamma rhythms in brain, the same has not been rigorously tested in humans. In this case-control study, we recruited a large population (N = 244; 106 females) of elderly (>49 years) subjects from the community, who viewed large gratings that induced strong gamma oscillations in their electroencephalogram (EEG). These subjects were classified as healthy (N = 227), mild cognitively impaired (MCI; N = 12), or AD (N = 5) based on clinical history and Clinical Dementia Rating scores. Surprisingly, stimulus-induced gamma rhythms, but not alpha or steady-state visually evoked responses, were significantly lower in MCI/AD subjects compared to their age- and gender-matched controls. This reduction was not due to differences in eye movements or baseline power. Our results suggest that gamma could be used as a potential screening tool for MCI/AD in humans.

## Introduction

Alzheimer's disease (AD) is a predominant cause of dementia (decline in cognitive abilities) of old age and substantially contributes to the socioeconomic burden in the geriatric population, necessitating early diagnosis. Advances in our understanding of cellular pathology of AD in rodent models and its link to gamma rhythms in brain have spurred interest to investigate diagnostic and therapeutic potential of gamma rhythms in AD and other forms of dementia (*Mably and Colgin, 2018*; *Palop and Mucke, 2016*).

Gamma rhythms are narrow-band oscillations in brain's electrical activity with center frequency occupying ~30–80 Hz frequency range (*Gray et al., 1989*). These are suggested to be generated from excitatory-inhibitory interactions of pyramidal cell-interneuron networks (*Buzsáki and Wang, 2012*) involving parvalbumin (PV) and somatostatin (SOM) interneurons (*Cardin et al., 2009*; *Sohal et al., 2009*; *Veit et al., 2017*). These have been proposed to be involved in certain cognitive functions like feature binding (*Gray et al., 1989*), attention (*Chalk et al., 2010*; *Fries et al., 2001*; *Gregoriou et al., 2009*), and working memory (*Pesaran et al., 2002*).

Some studies have reported abnormalities in gamma linked to interneuron dysfunction in AD. For example, *Verret et al., 2012* reported PV interneuron dysfunction in parietal cortex of AD patients and hAPP mice (models of AD). They found aberrant gamma activity in parietal cortex in such mice.

**eLife digest** Alzheimer's disease is one of the most common forms of dementia, characterised by declining memory and thinking skills, and behavioural changes that worsen over time. It affects millions of people worldwide, mostly in older age, and yet early indicators of the disease are lacking. Most cases are only diagnosed once a person's brain function becomes noticeably impaired, even though known biological changes underpin the disease. Detecting Alzheimer's disease early could aid diagnosis and enable early intervention, while also improving the chances of finding treatments to halt or reverse the disease.

Currently, brain function is measured by performing cognitive tests, such as remembering a set of words, imaging the brain with MRIs or CT scans, and blood or spinal fluid tests. Many of these tests can be invasive and expensive, so researchers are exploring whether measuring oscillations in the brain's electrical activity can be a non-invasive and chepaer way of testing brain function. Gamma oscillations are rhythmic signals, thought to be involved in attention and working memory. Animals used to study Alzheimer's disease have shown some abnormalities in gamma oscillations, and studies of healthy humans have also observed a decline in the strength and frequency of these oscillations with age. These findings have spurred an interest in understanding the link between gamma oscillations and AD in humans.

To investigate this link, Murty et al. measured patterns of brain activity in elderly people chosen from the community using electrodes placed on their scalps (a technique called electroencephalography). These participants watched certain images previously shown to elicit gamma oscillations. Participants who were later diagnosed with early Alzheimer's disease had weaker gamma oscillations than their cognitively healthy peers in the part of the brain that processes visual images.

These results build upon previous findings from animal research suggesting that gamma oscillations may be disrupted in early Alzheimer's disease. The work by Murty et al. could lead the way to new ways of diagnosing Alzheimer's disease, where early indicators are urgently needed.

Further, some recent studies have suggested therapeutic benefit of entraining brain oscillations in gamma range in rodent models of AD. For example, *Iaccarino et al., 2016* suggested that visual stimulation using light flickering at 40 Hz entrained neural activity at 40 Hz and correlated with decrease in Aβ amyloid load in visual cortices of 5XFAD, APP/PS1 mice models of AD. Based on such reports in rodents in both visual and auditory modalities, some investigators have suggested a paradigm termed GENUS (gamma-entrainment of neural activity using sensory stimuli) and have claimed to show neuro-protective effects in rodent models of AD (*Adaikkan et al., 2019*; *Martorell et al., 2019*).

Recent studies in human EEG (*Murty et al., 2020*; *Murty et al., 2018*) and MEG (*Pantazis et al., 2018*) have reported existence of two gamma rhythms (slow: ~20–34 Hz and fast: ~36–66 Hz) in visual cortex, elicited by Cartesian gratings. Age-related decline in power and/or frequency of these stimulus-induced gamma rhythms has been shown in cognitively healthy subjects (*Gaetz et al., 2012*; *Murty et al., 2020*). However, abnormalities in such stimulus-induced visual narrow-band gamma rhythms in human patients of mild cognitive impairment (MCI, a preclinical stage of dementia [*Albert et al., 2011*; *Petersen et al., 1999*; *Sosa et al., 2012*]) or AD have not been demonstrated till date.

We addressed this question in the present double-blind case-control EEG study involving a large cohort of elderly subjects (N = 244; 106 females, all aged >49 years) recruited from urban communities in Bangalore, India. These were classified as healthy (N = 227; see *Murty et al., 2020*), or suffering from MCI (N = 12) or AD (N = 5) based on clinical history and Clinical Dementia Rating (CDR; *Hughes et al., 1982*; *Morris, 1993*) scores. We studied narrow-band gamma rhythms induced by full-screen Cartesian gratings in all these subjects. We also examined steady-state visually evoked potentials (SSVEPs) at 32 Hz in a subset of these subjects (seven MCI and two AD subjects). We monitored eyes using an infrared eye tracker to rule out differences in gamma power due to potential differences in eye position or microsaccade rate (*Yuval-Greenberg et al., 2008*).

## Results

We presented achromatic full-screen static sinusoidal grating stimuli that varied in spatial frequency (1, 2, and 4 cpd) and orientation (0°, 45°, 90°, and 135°; *Figure 1A*) to a large cohort of elderly subjects (227 healthy, 12 MCI, and 5 AD subjects; see 'Materials and methods' for details of subject selection and classification). We first examined whether gamma power depended on orientation and spatial frequency. We have previously shown that gamma oscillations in EEG recorded from healthy young subjects have low orientation selectivity (*Murty et al., 2018*). Consistent with this, we found that stimulus-induced change in fast gamma power did not vary significantly across different orientations (*Figure 1—figure supplement 1*; two-way ANOVA with spatial frequency and orientation as factors; $F(3,2723) = 0.87$, p=0.46) in healthy subjects, with very low orientation selectivity (mean ± SEM: 0.02 ± 0.002; see 'Materials and methods' for details). Slow gamma varied with orientation ($F(3,2723) = 6.4$, p=0.0003); however, orientation selectivity calculated across the four orientations was small (mean ± SEM: 0.03 ± 0.002; see *Figure 1—figure supplement 1* for details). We therefore averaged all data across the four orientations.

As reported previously, gamma power varied across spatial frequencies as well (same two-way ANOVA as above; slow/fast gamma: $F(2,2723) = 31.1/50.7$, p=$4.6 \times 10^{-14}$/$2.4 \times 10^{-22}$). In particular, we found that the difference in gamma between cases and controls was more prominent at spatial frequencies of 2 and 4 cpd, so we used the data for these two spatial frequencies for main analysis. The 1 cpd condition, as well as the power averaged across all three spatial frequencies, gave similar, albeit slightly weaker, results (as shown in *Figure 2—figure supplement 3*).

Because the sample sizes were severely unbalanced between cases and controls, for each case subject, we selected controls who were age- (±1 year) and gender-matched and averaged their spectral data. All results shown here are based on pairwise comparison between cases and their

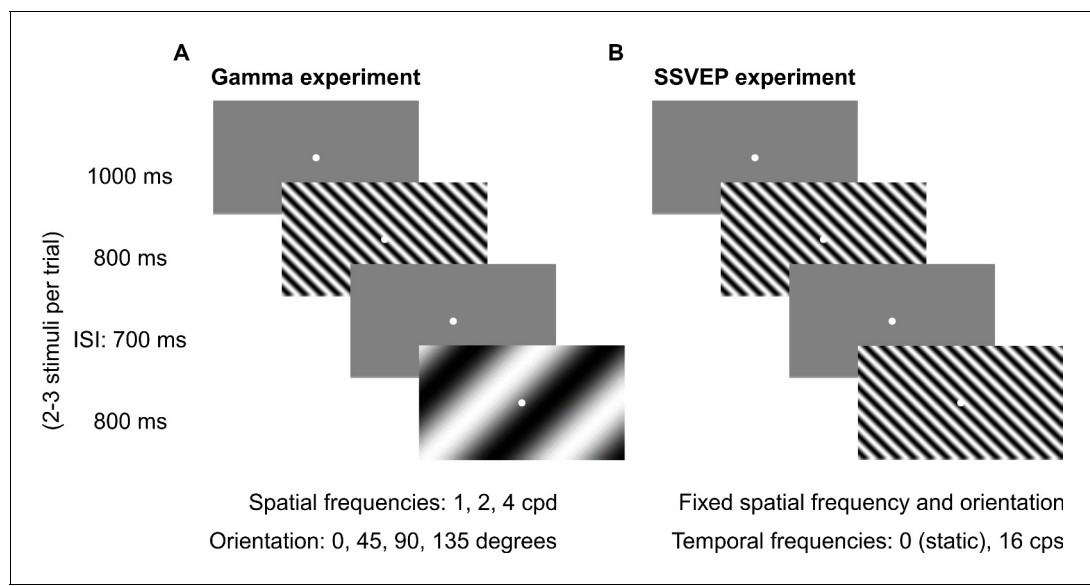

**Figure 1.** Fixation task. Every trial started with the onset of a fixation spot (0.1°) at the center of the screen on which the subjects had to maintain fixation. After an initial blank period of 1000 ms (gray screen), 2–3 stimuli were randomly shown for 800 ms. These consisted of sinusoidal luminance gratings presented full screen at full contrast. Inter-stimulus interval (ISI) was 700 ms. Each stimulus (of a particular combination of spatial frequency, temporal frequency, and orientation) was presented for a total of ~30–40 times according to the subjects' comfort and willingness, and is referred to as a 'stimulus repeat' in this paper unless otherwise stated. (**A**) Gamma experiment: static gratings (temporal frequency = 0 Hz) were presented at three spatial frequencies (SFs): 1, 2, and 4 cycles per degree (cpd) and four orientations: 0°, 45°, 90°, and 135°. This experiment lasted for ~25 min, with 1–2 short breaks (for 3–5 min) between blocks. (**B**) SSVEP experiment: gratings were randomly presented at a temporal frequency of 0 (static) or 16 cycles per second (cps). SF and orientation combination of gratings was fixed across the experiment. This was the combination that showed high change in slow and fast gamma power for each subject during preliminary analysis performed during the session. This experiment followed Gamma experiment during the same session and lasted for ~5 min completed in one block.

The online version of this article includes the following figure supplement(s) for figure 1:

**Figure supplement 1.** Slow and fast gamma for different orientations and spatial frequencies in healthy subjects.

averaged controls. Non-pairwise comparison (e.g., between 12 MCI and their 74 age- and gender-matched controls) yielded similar results.

## Change in gamma power, but not alpha suppression, was reduced in case group compared to control group

First, we examined how the two gamma rhythms differed in MCI subjects as compared to their healthy age- and gender-matched controls. We averaged spectral data for all analyzable bipolar electrodes (as described in *Murty et al., 2020*) from nine occipital and parieto-occipital pairs (marked in black enclosures in *Figure 2D*; see EEG data analysis subsection in 'Materials and methods') for each subject. We compared the change in power spectral densities (PSDs) for each MCI with the mean change in power of their corresponding age- and gender-matched controls. *Figure 2A* shows the median stimulus-induced change in PSDs for 12 MCIs (yellow) and their controls (dark orange; light shaded regions show ± SD of median after bootstrapping for 10,000 iterations). While both slow and fast gamma 'bumps' were conspicuously visible for MCI as well as control groups, power in both slow gamma (20–34 Hz) and fast gamma (36–66 Hz) ranges (but not alpha, 8–12 Hz range) was significantly lower in the MCI group compared to the control group (Kruskal-Wallis [K-W] test, significance as shown in *Figure 2A*). This could also be seen in the median time-frequency change in power spectrograms (baseline: −500–0 ms of stimulus onset) for cases and controls in *Figure 2B*. Change in band-limited power was significantly less for both gamma bands in the MCI group compared to the control group (*Figure 2C*; slow gamma: $\chi^2(23) = 4.09$, p=0.043; fast gamma: K-W test, $\chi^2(23) = 5.61$, p=0.018). However, alpha power was not significantly different ($\chi^2(23) = 0.33$, p=0.56). Results were similar when we combined both gamma bands to a single band (20–66 Hz; $\chi^2(23) = 5.08$, p=0.024) or used the 'traditional' gamma band (30–80 Hz; $\chi^2(23) = 5.34$, p=0.021). *Figure 2—figure supplement 1* shows the time-frequency change in power spectra of individual MCI subjects and the mean of their controls, sorted by increasing gamma power in the MCI subjects. Although there was substantial variability across subjects (also observed in *Figure 2C*), only 3/12 MCIs showed higher slow gamma power and only 2/12 MCIs showed higher fast gamma than controls.

We note that we have used very stringent conditions for computation of gamma power, similar to our previous work (*Murty et al., 2020*; *Murty et al., 2018*). For example, for all subjects, we used the same set of electrodes over which gamma was computed, as well as same time and frequency ranges. Further, we computed the total power within a band by simply summing the absolute power values within the band separately in baseline and stimulus periods and then taking a ratio. This estimate has larger contribution from lower frequencies in the band because of the power-law distribution of PSDs in baseline/stimulus periods. Consequently, if the traces are overlapping at lower frequencies within the band and diverge at higher frequencies, which was the case in the slow gamma range, the total change in power in the band may not be significantly different. Therefore, these results could be further improved by customizing the low frequency limit of the gamma band for each subject, as well as choosing only electrodes that show stronger gamma. For example, taking slow and fast gamma ranges as 26–34 Hz and 44–56 Hz improved the p-values (slow gamma: $\chi^2(23) = 7.69$, p=0.005; fast gamma: $\chi^2(23) = 8.34$, p=0.004). Although we have refrained from such customization here because we wanted to study the efficacy of a simple and subject-independent computational procedure, such data-driven subject specific optimization holds promise for improving the efficacy of a gamma-based biomarker.

*Figure 2D* shows the median scalp maps (see EEG data analysis subsection in 'Materials and methods') of change in band-limited power across 112 bipolar electrode pairs (shown as discs) for alpha, slow, and fast gamma bands. Stimulus-induced change in power across all three bands was most prominent in the nine electrode pairs as above. However, this change in power was less in the MCI group compared to the control group in slow and fast gamma bands (but not alpha band) as noted in *Figure 2C*.

For two MCI subjects (M1 and M4 as shown in *Figure 2—figure supplement 1*), slow gamma power was less than 0 dB. This could be because visual stimuli typically suppress power at low frequencies (<30 Hz), and if the visual stimulus does not induce sufficiently strong slow-gamma rhythm, there is an overall reduction in power between 20 and 35 Hz. However, our results remained consistent even when these two MCI subjects were removed from analysis ($\chi^2(19) = 4.81$, p=0.028, for

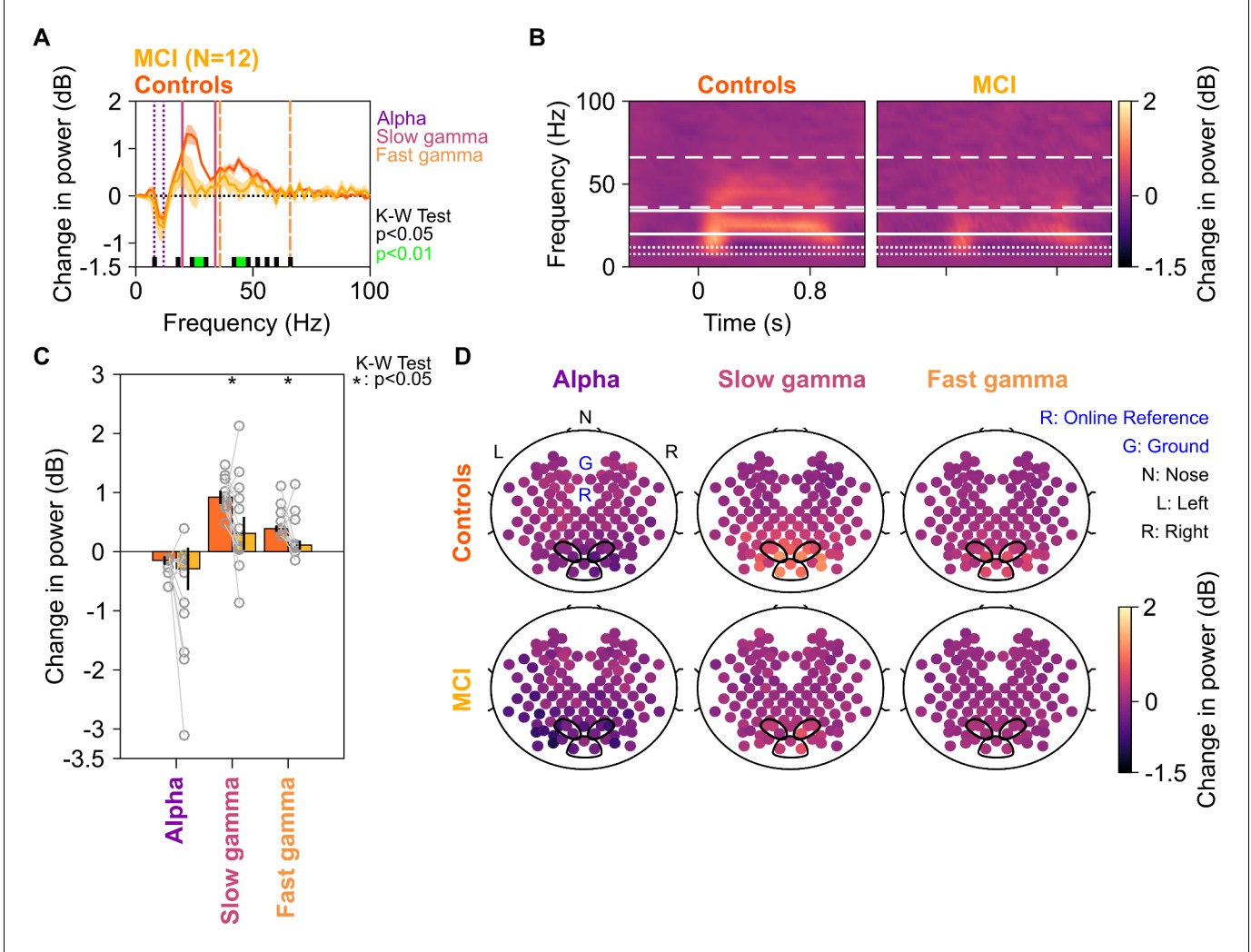

**Figure 2.** Alpha, slow, and fast gamma in mild cognitive impairments (MCIs) and controls. (**A**) Change in power spectral densities (PSD) for 12 MCI subjects and their respective controls. Solid traces indicate median PSD across 12 MCIs (yellow) and median of mean PSDs for 12 sets of healthy controls (dark orange). Shaded regions indicate ± SD from medians after bootstrapping over 10,000 iterations. Vertical lines represent alpha (8–12 Hz, violet), slow (20–34 Hz, pink), and fast gamma (36–66 Hz, orange). Colored bars at the bottom represent significance of differences in medians (K-W test, black: p=0.01–0.05, green: p<0.01; not corrected for multiple comparisons). (**B**) Median change in power spectrograms for MCIs (right) and controls (left). White horizontal lines represent alpha (dotted), slow gamma (solid), and fast gamma (dashed) bands. (**C**) Median change in power in alpha, slow gamma, and fast gamma bands for MCIs (yellow) and controls (dark orange). Data of individual MCI and mean for their respective controls are represented as gray circles. Error bars indicate ± SD from medians after bootstrapping over 10,000 iterations. Black asterisks represent significance of differences in medians (K-W test, p<0.05, not Bonferroni-corrected). (**D**) Average scalp maps of 112 bipolar electrodes (disks) for cases (bottom row) and controls (top row) for alpha (left), slow gamma (middle), and fast gamma (right). Color of disks represents change in power in respective frequency bands. Electrode groups used for calculation of band-limited power are enclosed in black.

The online version of this article includes the following figure supplement(s) for figure 2:

**Figure supplement 1.** Time-frequency change in power spectrograms for MCI compared to their corresponding controls.

**Figure supplement 2.** Alpha, slow, and fast gamma in MCI subjects and fewer controls.

**Figure supplement 3.** Alpha, slow, and fast gamma in MCI subjects and controls across different spatial frequencies.

**Figure supplement 4.** Evoked potentials in MCIs and healthy controls.

slow gamma between 26 and 34 Hz). Similarly, our results did not change when we removed one MCI subject (subject M5) who had negative fast gamma power ($\chi^2(21) = 4.56$, p=0.032).

Our study had many control subjects for each case subject (range 4–19). To rule out the possibility that our results were influenced by this imbalance, we first ordered the control subjects based on the difference in experiment date from the case subject and then chose only the first N control

subjects. We tried different values of N and found that our results remained consistent in all cases, with less slow and fast gamma power in cases vs controls. For example, *Figure 2—figure supplement 2* shows the results for N = 1 (a single control for each case).

Although the error bars shown in *Figure 2C* appear smaller for controls than cases, that is only because each data point for the control group is already an average across many subjects. When a single control subject was used per case (*Figure 2—figure supplement 2*), the error bars were comparable (standard error of the medians: 0.33, 0.20, and 0.34 for controls vs 0.28, 0.08, and 0.36 for cases, for slow gamma, fast gamma, and alpha, respectively). Similarly, if we pooled all the control subjects in one group without averaging (N = 74 controls vs 12 MCI subjects), the standard deviations of slow gamma, fast gamma, and alpha power were 0.94, 0.72, and 0.67 for controls and 0.79, 0.36, and 1.00 for cases. Therefore, the variability in power was comparable in the two groups.

*Figure 2—figure supplement 3* shows the results for individual spatial frequencies as well as after pooling all three spatial frequencies. MCI subjects had less slow and fast gamma than controls at all spatial frequencies, although the effect was stronger at spatial frequencies of 2 and 4 cpd.

We also tested whether the event-related potentials (ERPs) varied across MCIs and healthy controls. Consistent with literature, we noticed three prominent peaks in the ERPs for these subjects: P1, N1, and P2 (*Figure 2—figure supplement 4*). The ERPs were not different between MCIs and controls (*Figure 2—figure supplement 4*, panel A). Specifically, we did not find any significant difference between P1/N1/P2 peak amplitude (*Figure 2—figure supplement 4*, panel B). These analyses indicate that the differences that we observed for MCIs and controls were limited only to slow and fast gamma power.

*Figure 3* shows the same results for the five AD subjects in our cohort. We interpret these results with caution, since the number of subjects is small (although the study would have benefitted from a larger sample size for both MCI and AD categories, it was not possible to increase the sample size due to the COVID-19 pandemic). Nonetheless, we observed a strong reduction in the slow and fast gamma bands (*Figure 3A and B*), which was significant for both slow gamma ($\chi^2(9) = 4.81$ p=0.028) and fast gamma ($\chi^2(9) = 3.94$, p=0.047), but not alpha ($\chi^2(9) = 0.01$, p=0.92). Data from individual AD subjects and their controls are shown in *Figure 3—figure supplement 1*. All five AD subjects had less slow gamma power, and only one AD subject (A5) had more fast gamma power relative to the controls. Further, similar to MCI subjects, AD subjects had less slow and fast gamma than controls at all spatial frequencies, although the effect was stronger at spatial frequencies of 2 and 4 cpd (*Figure 3—figure supplement 2*).

Because four out of five AD subjects had only mild AD (CDR = 1), we also tested the results after combining the MCI and AD data sets (*Figure 3—figure supplement 3*). While slow gamma ($\chi^2(33) = 9.51$, p=0.002) and fast gamma ($\chi^2(33) = 8.88$, p=0.003) power were strongly reduced in cases vs controls, there was no difference in alpha power ($\chi^2(33) = 0.25$, p=0.61) or frequencies higher than ~65 Hz.

We further tested the dependence of power in gamma/alpha bands on CDR score using a linear regression model (see Regression analysis section in 'Materials and methods') while accounting for age and gender. In the matched condition in which data from all the age- and gender-matched control subjects for each case were averaged (yielding 17 cases and 17 controls), the coefficient for CDR was significantly negative for both gamma bands ($\beta_{CDR}$ = –0.57/–0.25, p=0.0017/0.0063 for slow/fast gamma). Results were similar for the unmatched condition, in which all the controls were considered separately (112 healthy, 12 MCI, and 6 AD subjects), albeit the CDR slope ($\beta_{CDR}$) was significantly negative only for slow gamma ($\beta_{CDR}$ = –0.62/–0.29, p=0.0071/0.0702 for slow/fast gamma). On the other hand, similar to the results in previous analyses, alpha power did not depend on CDR ($\beta_{CDR}$ = –0.30/–0.30, p=0.14/0.21 for matched/unmatched conditions).

We have previously shown that gamma power decreases with age in healthy elderly, and females have more gamma than males (*Murty et al., 2020*). Consistent with these results, we found that coefficient for age was significantly negative ($\beta_{AGE}$ = –0.018/–0.0094, p=0.014/0.06 for slow/fast gamma), while coefficient for gender was significantly positive ($\beta_{GENDER}$ = 0.33/0.35, p=0.0070/$3.15 \times 10^{-5}$ for slow/fast gamma) when the regression analysis was performed on the full set of healthy subjects (N = 227). However, when the regression analysis was performed with cases and controls as described above, the coefficients were not significant (matched: $\beta_{AGE}$ = 0.0042/$4.13 \times 10^{-5}$, p=0.73/0.99 and $\beta_{GENDER}$ = –0.036/0.11, p=0.87/0.36 for slow/fast gamma; unmatched:

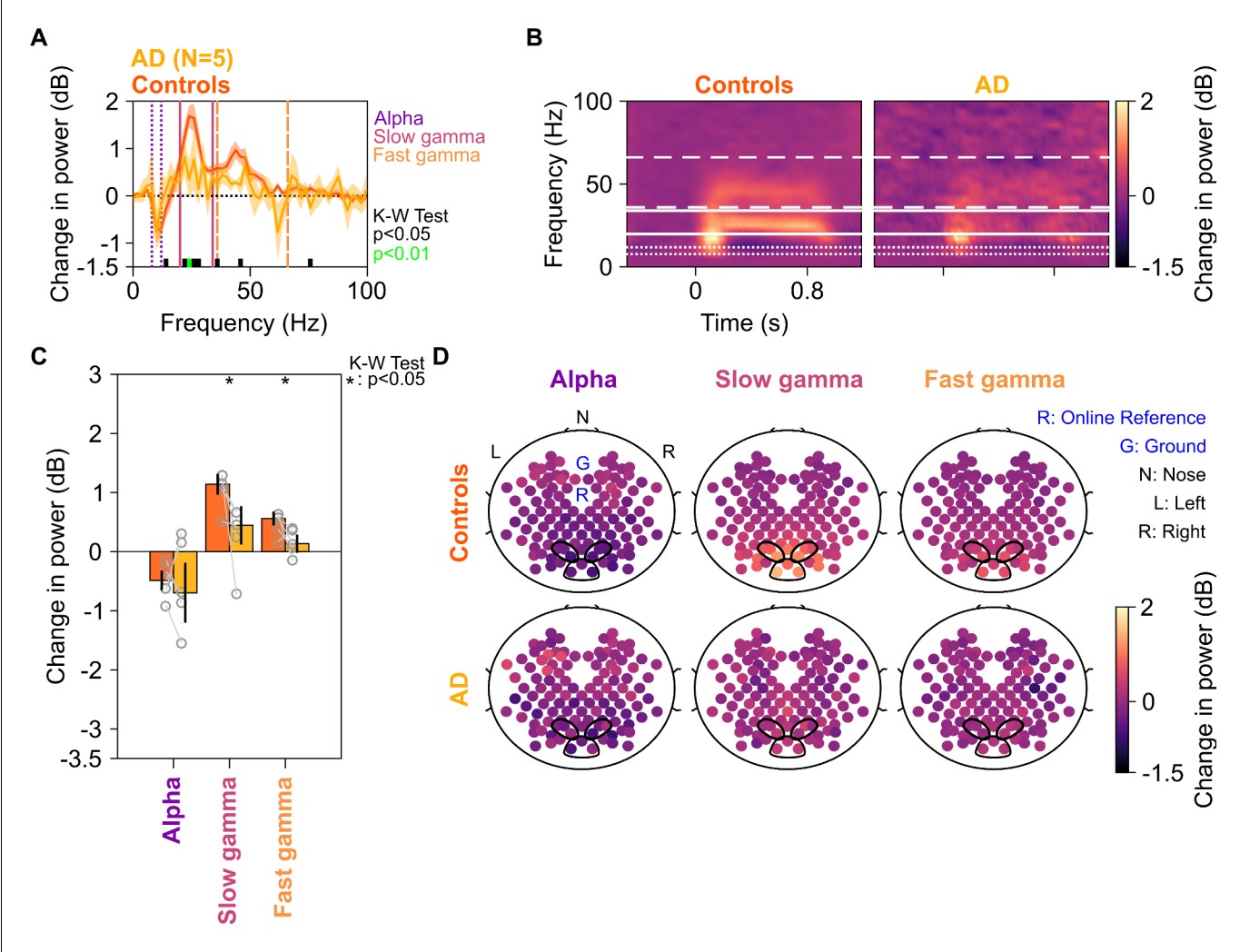

**Figure 3.** Alpha, slow, and fast gamma in AD subjects and controls. Same format as in *Figure 2*, but for five AD subjects and their respective controls. The online version of this article includes the following figure supplement(s) for figure 3:

**Figure supplement 1.** Time-frequency change in power spectrograms for AD compared to their corresponding controls.
**Figure supplement 2.** Alpha, slow, and fast gamma in AD subjects and controls across different spatial frequencies.
**Figure supplement 3.** Alpha, slow, and fast gamma in all cases (MCIs and ADs) and controls.

$\beta_{AGE}$ = −0.0071/−0.0009, p=0.51/0.91 and $\beta_{GENDER}$ = 0.17/0.19, p=0.30/0.10 for slow/fast gamma). This also suggests that CDR is a stronger predictor of gamma power than age or gender.

These results suggest that the alternate hypothesis (gamma power in controls was greater than cases) was more likely than the null hypothesis (controls and cases had comparable gamma power). We quantified this by estimating the Bayes factor (BF), which is the ratio of the marginal likelihood of the alternate hypothesis and the null hypotheses, given the data that we observed (see 'Materials and methods for details). For the MCI group (N = 12), BF computed using single-tailed paired t-test was ~1.89 for both slow and fast gamma. However, as before, choosing a more 'sensitive' range for slow gamma (26–34 Hz) and fast gamma (44–56 Hz) improved the BF to 3.34 and 4.60, respectively, suggesting substantial evidence for the alternate hypothesis over the null hypothesis. For the AD group (N = 5), BF was 2.85 for slow gamma and 1.83 for fast gamma, suggesting weak evidence, which did not improve substantially when more sensitive ranges were used. However, when both MCI and AD groups were combined, BF increased to 13.70 for slow gamma (26–34 Hz) and 8.60 for fast gamma (44–56 Hz), further strengthening the evidence in favor of alternate hypothesis. On the other hand, evidence for alpha band power was in favor of the null hypothesis

(BF was 0.088 when only MCIs were considered, 0.29 for ADs, and 0.077 when both MCI and AD subjects were considered).

## Difference in gamma power was not due to differences in eye position or movement

Previous studies have correlated increases in gamma power with occurrence of small involuntary eye movements called microsaccades (*Yuval-Greenberg et al., 2008*). These have been described in previous literature using plots called 'main sequence,' which show peak velocity on ordinate and maximum velocity on abscissa on a log-log scale. These plots reveal the ballistic nature of microsaccades, that is, the initial velocity and maximum displacement of the eye in the visual field are correlated during microsaccadic movements (*Engbert, 2006*). We compared eye data between MCI/AD subjects and their respective controls and found comparable eye positions and microsaccade profiles (*Figure 4A*), similar main sequence (*Figure 4B*), and similar pupillary reactivity (*Figure 4C*) to stimulus presentation (measured as coefficient of variation of pupil diameter across time; see *Murty et al., 2020*). Further, the trends described in *Figures 2* and *3* did not change qualitatively when we reanalyzed the data after removing stimulus repeats containing microsaccades (see 'Materials and methods') from analysis (*Figure 4—figure supplement 1*), although these did not reach significance due to lesser number of trials (~45% of original analysis) and fewer subjects compared to the original analysis (see figure legend for details). Similarly, the trends held true when we reanalyzed only those repeats that had at least one microsaccade (*Figure 4—figure supplement 2*). These results indicate that the trends described in *Figure 2* are independent of the presence or absence of microsaccades.

## Difference in gamma power was not due to differences in baseline power

We also tested if the trends described in *Figures 2* and *3* were seen for absolute band-limited power in the baseline condition. The cases (MCIs or ADs) and their respective controls had comparable PSDs (*Figure 5*) and slopes of PSDs (*Figure 5—figure supplement 1*) in the baseline condition. Further, baseline power in alpha, slow gamma, and fast gamma frequency ranges did not differ significantly between cases and controls (K-W test; for MCIs: $\chi^2(23)$ = 0.48/0/0, p=0.49/0.95/1 for alpha/slow/fast gamma, respectively; for AD: $\chi^2(9)$ = 0.27/0.53/0.27, p=0.60/0.46/0.60 for alpha/slow/fast gamma, respectively). Thus, we concluded that the trends described in *Figures 2* and *3* were specific to stimulus-induced change in slow/fast gamma power and did not depend on baseline absolute power or slopes of PSDs.

## SSVEP power at 32 Hz was comparable in case and control groups

We next tested whether power of SSVEPs in gamma range also decreased in the MCI group as compared to the control group. We tested for SSVEPs at 32 Hz by presenting full-screen gratings that phase-reversed at 16 Hz (as described in *Figure 1B*). Ten of the 12 MCIs participated in this study, out of which data of only seven could be analyzed (data from three MCIs were discarded due to noise, see 'Materials and methods' for details). *Figure 6A and B* show median change in power spectral density plots (in 250–750 ms window of stimulus onset) and change in power time-frequency spectrograms (from a baseline period of −500–0 ms of stimulus onset) for these seven MCIs and their respective age- and gender-matched controls (as done for analyses in *Figure 2A and B*). *Figure 6C and D* show bar plots and scalp maps for 112 bipolar electrodes for change in SSVEP power at 32 Hz (during 250–750 ms of stimulus onset from a baseline of −500–0 ms, same as in *Figure 2C and D*) for control and MCI groups, respectively. We did not observe any reduction in SSVEP power at 32 Hz in the MCI group as compared to the control group (K-W Test, significance at each frequency of the change in power spectra is shown in *Figure 6A*; significance at 32 Hz: $\chi^2(13)$ = 0.04, p=0.85). For comparison, we reanalyzed data for slow and fast gamma power for the Gamma experiment (as in *Figure 2*) with the same set of seven MCI subjects and their respective controls. MCIs had less slow and fast gamma power compared to controls as seen in *Figure 6—figure supplement 1* (same format as *Figures 2* and *3*), although these trends did not reach significance due to small sample size (K-W test, $\chi^2(13)$ = 0.49/2.56, p=0.48/0.11 for slow/fast gamma, respectively). As before, choosing the more sensitive slow-gamma frequency range between

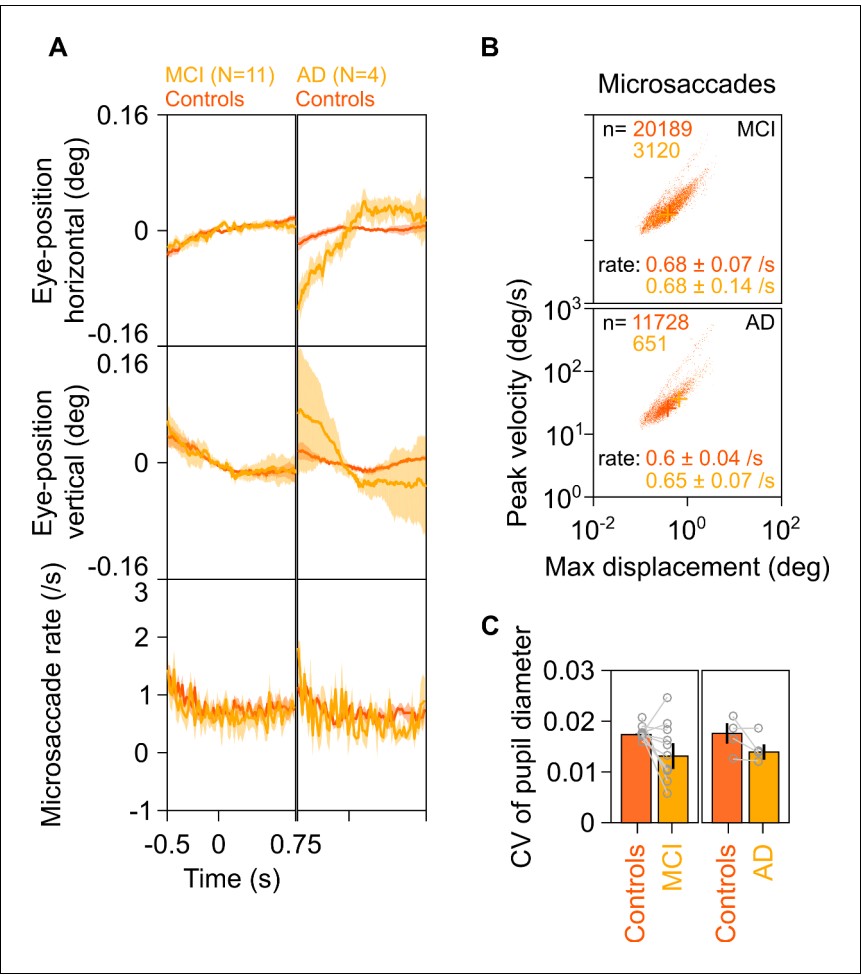

**Figure 4.** Eye position, microsaccades, and pupillary reactivity for healthy/MCI/AD subjects. (**A**) Left column: Eye position in horizontal (top row) and vertical (middle row) directions; and histogram showing microsaccade rate (bottom row) vs time (−0.5–0.75 s of stimulus onset) for 11 MCI cases (yellow) and their respective healthy controls (dark orange). Solid traces indicate medians, shaded patches represent ± SD of median after bootstrapping over 10,000 samples. Right column: Same plots for four AD cases and their healthy controls. Eye position did not vary significantly across time between MCI/AD and control subjects except in the case of AD vs controls, where it varied slightly (but within ±0.1˚). (**B**) Main sequence plots showing peak velocity and maximum displacement of all microsaccades (number indicated on top) extracted from 11 MCI (top row), 4 AD (bottom row) subjects indicated in yellow, and their corresponding healthy controls (dark orange). Average microsaccade rate (median ± SD of median of 10,000 bootstrapped samples) across all subjects for each group is also indicated at the bottom of the panels. MCI/AD cases had similar microsaccade rates (also seen in panel A) and main sequence plots compared to their healthy controls. (**C**) Bar plots showing coefficient of variation of pupil diameter (reactivity of pupil to stimulus presentation; see *Murty et al., 2020*) for 11 MCI (left), 4 AD (right), and their corresponding healthy controls. Data for individual MCIs and average across respective controls is represented by gray circles. Height of bars indicate medians and error bars indicate ± SD of median of 10,000 bootstrapped samples. We did not find any significant difference between the MCI/AD and control groups in pupil reactivity (K-W test, MCI vs controls: $\chi^2(21) = 3.76$, p=0.052; AD vs controls: $\chi^2(7) = 0.75$, p=0.39).

The online version of this article includes the following figure supplement(s) for figure 4:

**Figure supplement 1.** Spectral analyses for trials containing no microsaccades.
**Figure supplement 2.** Spectral analyses for trials containing microsaccades.

26 and 34 Hz yielded a significant difference between the two groups (K-W test, $\chi^2(13) = 3.93$, p=0.047). Alpha followed a similar insignificant trend as in *Figure 2* ($\chi^2(13) = 0$, p=0.949). Further, adding the two AD subjects yielded significant differences in both slow and fast gamma power (K-W test, slow gamma between 26 and 34 Hz in Gamma experiment: $\chi^2(17) = 4.31$, p=0.038; fast

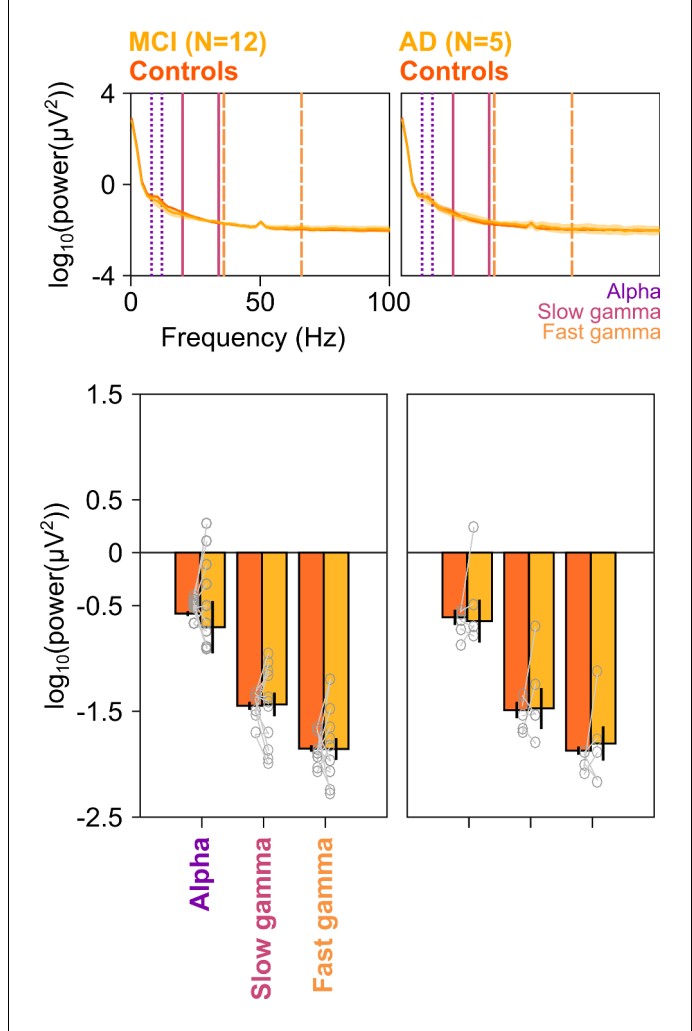

**Figure 5.** Baseline PSDs and alpha/slow/fast gamma power in cases and healthy controls. Left column: Baseline PSDs (top row) and bar plots (bottom row) showing baseline absolute power (calculated in −500–0 ms of stimulus onset) for each of the 12 MCIs and corresponding healthy controls in alpha, slow gamma, and fast gamma bands. Same format as in *Figure 2A and C*. Data for individual MCIs and averages of corresponding control subjects are shown in gray circles. Corresponding analyses for five AD subjects are shown in right column. None of the differences in MCI and AD groups (compared to controls) were significant (see Results section).
The online version of this article includes the following figure supplement(s) for figure 5:

**Figure supplement 1.** Baseline slopes in cases and healthy controls.

gamma: $\chi^2(17) = 4.69$, p=0.03), but not for SSVEP power (for change in power at 32 Hz in SSVEP experiment: $\chi^2(17) = 0.05$, p=0.82).

Trends for SSVEPs were not different when we performed time-frequency analysis on trial-averaged time-amplitude waveforms for each subject (*Figure 6—figure supplement 2A–C*), instead for averaging time-frequency spectra of individual repeats for each subject, as done above for Gamma and SSVEP analyses. Further, stimulus onset-related responses (0–250 ms) were comparable between MCIs and their respective controls (*Figure 6—figure supplement 2D*). The RMS amplitude, peak amplitude, and time of the peak of the stimulus-onset response also did not differ among MCI subjects and their respective controls (*Figure 6—figure supplement 2D–E*). To conclude, change in SSVEP power at 32 Hz for cases was comparable to that of their controls, like alpha but unlike slow and fast gamma oscillations.

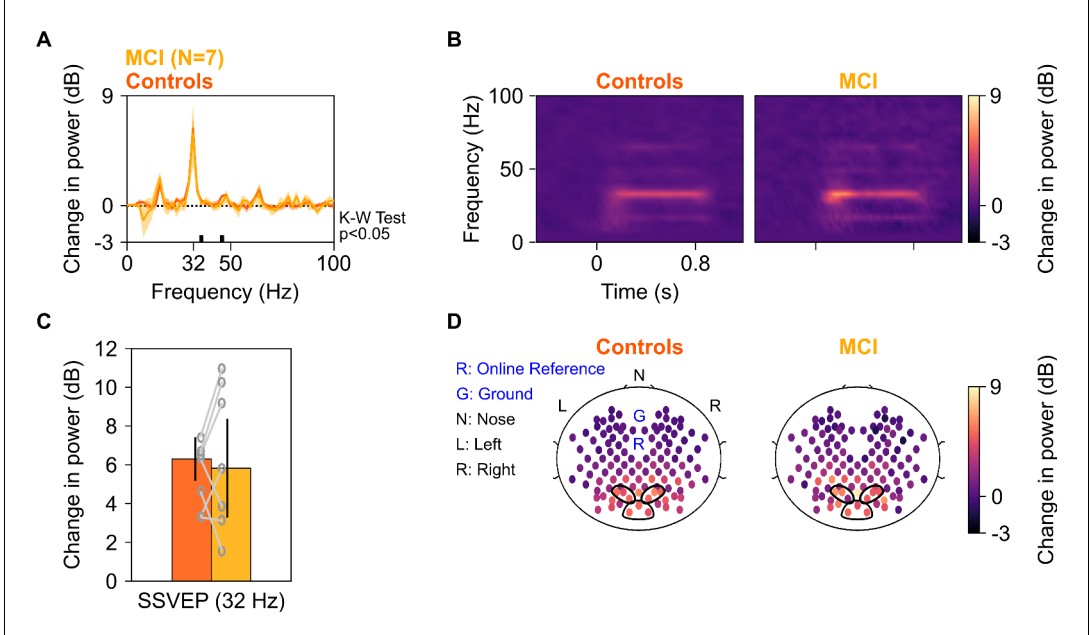

**Figure 6.** SSVEP at 32 Hz in MCIs and controls. Median change in power spectral density (PSD) (solid traces in **A**) and median change in power spectrograms in (**B**) for seven MCIs and their respective controls. Shaded regions in (**A**) indicate ± SD from medians after bootstrapping over 10,000 iterations. (**C**) Median change in SSVEP power at 32 Hz for MCIs (yellow) and controls (dark orange). Error bars indicate ± SD from medians after bootstrapping over 10,000 iterations. (**D**) Median scalp maps of 112 bipolar electrodes (disks) for MCIs (bottom row) and controls (top row) for change in power at 32 Hz. Same format as in *Figure 2D*.

The online version of this article includes the following figure supplement(s) for figure 6:

**Figure supplement 1.** Alpha, slow, and fast gamma in seven MCIs and their healthy controls used for SSVEP analysis.

**Figure supplement 2.** Analysis of trial-averaged time-amplitude waveforms from SSVEP experiment.

## Discussion

Stimulus-induced change in power of both narrow-band slow and fast gamma oscillations reduced in elderly subjects with MCI and AD compared to their age- and gender-matched healthy controls. We removed or ruled out possible biases due to peripheral ocular factors or overall baseline noise of the PSDs to further strengthen the results. In contrast to gamma, we did not find any significant reduction in stimulus-induced alpha suppression, ERP, or SSVEP at 32 Hz in cases.

Previous studies have suggested abnormalities in spontaneous and evoked activity in gamma band in various disorders. Some studies have also explored abnormalities in other oscillatory bands such as alpha as well as functional connectivity across brain regions in such disorders. Examples include autism (*An et al., 2018*; *Uhlhaas and Singer, 2007*), schizophrenia (*Hirano et al., 2015*; *Uhlhaas and Singer, 2010*), and AD and other dementias (*Herrmann and Demiralp, 2005*; *Jeong, 2004*; *Pievani et al., 2011*). However, this is the first study to our knowledge that found abnormalities in visual narrow-band gamma oscillations elicited by Cartesian gratings in human EEG in MCI and AD.

Previously, *van Deursen et al., 2008* examined neural activity in gamma frequency range and found that gamma activity increased in subjects with MCI/AD, as opposed to their initially proposed hypothesis. Our findings are different from theirs, probably due to the differences in task settings: we had tested for visual gamma rhythms induced by gratings, whereas they had tested for resting-state gamma as well as gamma activity for multiple-colored objects randomly moving on a computer screen (taken from screensavers). Whether our results hold true for drifting (moving) and chromatic gratings remains to be tested in future studies.

## Potential of gamma rhythms as electrophysiological markers for MCI

Our sample is a representative of urban population in India as we adopted community-based sampling instead of hospital-based sampling. Importantly, out of the 257 subjects that we collected data from (247 used for analysis plus 10 subjects whose data was noisy and thus rejected, see 'Materials and methods'), there were 13 MCIs (~5%) and 6 AD subjects (2.3%). These figures match closely to the previously reported prevalence of MCI and AD in India (*Kalaria et al., 2008*; *Mathuranath et al., 2012*; *Sosa et al., 2012*). Within the recruited sample, most cases (~70%) had MCI, a condition that is conceptualized as intermediate stage between normal aging and AD. Criteria used to diagnose MCI are not strong and hence there is a need for a valid biomarker. Our study highlights the potential use of gamma oscillations in EEG in that direction.

Moreover, we had limited our analyses to sensor (electrode) level instead of reconstructing the neural sources and performing analyses at that level. This has allowed us to present a screening technique that is easy to replicate in a clinical setting. Furthermore, as our metrics were derived directly from neural activity, these could serve as a more objective assessment of the clinical status of the individual. Future studies should examine if different other stimulus paradigms like annular gratings (*Muthukumaraswamy and Singh, 2013*) and drifting gratings (*Orekhova et al., 2015*) could add to the robust evidence that is presented in this study.

## Possible correlations of gamma power with neuroanatomy/physiology in AD

Gamma rhythms are generated by excitatory-inhibitory interactions in the brain (*Buzsáki and Wang, 2012*). These interactions could be influenced by many structural factors (*Buzsáki et al., 2013*) that could get abnormal in AD (such as cortical thinning and atrophy; see *Dickerson et al., 2009*; *Serrano-Pozo et al., 2011*). However, how such structural derailments influence gamma recorded over scalp is unknown. A few studies in MEG had reported significant positive correlations between gamma frequency and cortical thickness as well as volume of cuneus (*Gaetz et al., 2012*) and thickness of pericalcarine area (*Muthukumaraswamy et al., 2010*). However, such results could not be replicated in later studies (*Cousijn et al., 2014*) and have been shown to be confounded by age (*Robson et al., 2015*). Age is as a common factor that influences both macroscopic structure (*Lemaitre et al., 2012*; *Salat et al., 2004*; *van Pelt et al., 2018*) as well as gamma power and frequency (*Murty et al., 2020*).

Our main aim in this study was to examine the potential of gamma as a biomarker. However, as structural changes in AD brain are more evident and drastic compared to cognitively healthy aging (*Dickerson et al., 2009*; *Serrano-Pozo et al., 2011*), future studies should examine correlations of gamma power/frequency and macroscopic structure such as cortical thickness in healthy/MCI/AD subjects while controlling for age. Moreover, as gamma rhythms have been correlated with many higher level cognitive functions such as attention, working memory, etc. (see Introduction), attempts must be made to extend and validate these findings in clinical populations, such as AD.

## Comparison of SSVEP trends with findings in previous studies

Some investigators have suggested neuroprotective effects of entraining neural oscillations using flickering light/sound at 40 Hz (analogous to our SSVEP paradigm) in rodent models of AD (*Adaikkan et al., 2019*; *Martorell et al., 2019*; *Thomson, 2018*). Although we did not find any significant trend for SSVEP at 32 Hz in cases compared to controls (unlike our observations with narrow-band gamma), we cannot directly compare the results from abovementioned studies with our study for several reasons. First, the frequency of entrainment was different (40 Hz vs 32 Hz). Second, the model organism was different (human vs rodent). Finally, we did not measure any cognitive or pathological outcome of the visual stimulation. Indeed, while the previous studies have focused on therapeutic aspect of flicking stimulation, we only studied its potential for diagnosis. It is possible that entrainment of neural oscillations to visual stimulation in gamma frequency range gets deranged only in advanced stages of AD (as in the rodent models of *Iaccarino et al., 2016*). It may thus have therapeutic benefit but may not reflect as abnormal on testing early on (as in our case). Further, as discussed in the Methods, SSVEP study was always done at the end of the experiment in our study and the total number of stimulus repeats were much less than the gamma study. Nonetheless, the differences in trends for gratings and SSVEP evoked gamma presented in this study suggest that

these two phenomena might be sub-served by different, yet unknown mechanisms. These differences have to be borne in mind while designing screening/therapeutic tools for MCI and AD.

## Conclusions

Stimulus-induced change in visual narrow-band gamma power has the potential to be a simple, low-cost, easy to replicate, and objective biomarker for screening of MCI and AD. How this gamma-based biomarker compares against other methods used in diagnosis (MRI, PET, cognitive tests, etc.), whether addition of this biomarker to other standard methods improves the overall diagnosis, and the specificity of this biomarker for AD compared to other causes of dementia remain open questions that will require further research.

# Materials and methods

## Key resources table

| Reagent type (species) or resource | Designation | Source or reference | Identifiers | Additional information |
|---|---|---|---|---|
| Software, algorithm | Chronux toolbox | chronux.org | RRID:SCR_005547 | - |
| Software, algorithm | EEGLAB toolbox | https://sccn.ucsd.edu/eeglab/index.php | RRID:SCR_007292 | - |

## Subjects

We recruited 257 elderly subjects (109 females) aged 50–92 years from the Tata Longitudinal Study of Aging (TLSA) cohort from urban communities in Bangalore between July 2016 and July 2019. They were clinically diagnosed by psychiatrists (authors BN/AML) and/or a neurologist (author MJ) as cognitively healthy (N = 236) or suffering from MCI (N = 15) or AD (N = 6) through clinical history and a semi-structured clinical interview (Clinical Dementia Rating; see *Table 1*). Five out of the six AD subjects were directly referred to the study by the neurologist. Diagnosis of all MCI/AD subjects

**Table 1.** Demographic and clinical details for subjects.

|  | Healthy | MCI | AD |
|---|---|---|---|
| *Demographic details* | | | |
| Number of subjects (no. of females in parentheses) | | | |
| Total recruited | 236 (104) | 15 (3) | 6 (2) |
| Total analyzed | 227 (101) | 12 (3) | 5 (2) |
| Age (in years, for analyzed subjects) | | | |
| Range (min-max) | 50–88 | 51–81 | 60–79 |
| Mean ± SD | 66.8 ± 8.2 | 71.4 ± 9.3 | 68.8 ± 7.7 |
| *Diagnostic criteria* | | | |
| Subjective memory complaint | Present/absent | Present | Present |
| General cognitive function* | Preserved | Preserved | Reduced |
| IADL | </=0.5 | </=0.5 | >0.5 |
| CDR | = 0 | = 0.5 | >0.5 |
| HMSE** | >27 | - | - |
| *Clinical scores of analyzed subjects* (no. of subjects in parentheses) | | | |
| CDR | 0 (204) | 0.5 (12) | 1 (4), 3 (1) |
| HMSE (mean ± SD) | 30.3 ± 1 (216) | 29.6 ± 1.5 (12) | 23 ± 2.7 (5) |

MCI: mild cognitive impairment; AD: Alzheimer's disease; IADL: Instrumental Activities of Daily Living (*Mathuranath et al., 2005*); CDR: Clinical Dementia Rating (*Hughes et al., 1982*; *Morris, 1993*); HMSE: Hindi Mental State Examination (*Ganguli et al., 1995*).

*Based on clinician's assessment.

**HMSE was used as a diagnostic criterion only if CDR score was unavailable and clinical testing did not indicate any sign of dementia.

was further reviewed by a panel of four experts for consensus (see Appendix; criteria used by the panel are given in Tables 1–3 in *Supplementary file 1*), who reclassified two MCI subjects as healthy. Data from these two subjects was not used for further analysis. Subjects went through the experiments only once. However, there were a few subjects who had undergone the experiments more than once (annually as part of a different longitudinal study). For such subjects, we used only data from the first year. However, in the first year of study, four subjects did not have eye data and data of one participant was noisy. Further, one participant was diagnosed as MCI in the second year. Hence, for these six subjects, data from the first year was discarded and data from the second year was used instead. From the rest, we discarded data of ten subjects due to noise (nine healthy and one MCI; see Artifact Rejection subsection). Finally, we discarded one AD patient (male, aged 92 years) as he did not have any healthy age- and gender-matched control. We were thus left with 227/12/5 (females: 101/3/2) healthy/MCI/AD subjects for analysis. For the purpose of this study, we called the MCI/AD subjects as cases and their respective age- and gender-matched healthy subjects as controls. Since the subjects were directly recruited from the community based on advertisements without any prior knowledge of their clinical status (which was determined during the study itself and revealed to the experimenters after the EEG recordings were over), no explicit power calculation was done.

All subjects reported normal or corrected-to-normal vision and were instructed to wear spectacles if prescribed earlier. They participated in the study voluntarily and were monetarily compensated for their time and effort. We obtained informed consent from all subjects before the experiment. The Institute Human Ethics Committees of Indian Institute of Science, NIMHANS, and MS Ramaiah Hospital, Bangalore approved all procedures. This article is in compliance with the STROBE statement (*von Elm et al., 2007*).

## Experimental setup and task

Experimental setup, EEG recordings, and analysis were same as what we had described in our previous study (*Murty et al., 2020*). Briefly, we recorded raw EEG signals from 64 active electrodes using BrainAmp DC (Brain Products GmbH, Germany) according to the international 10–10 system, referenced online at FCz. We filtered raw signals online between 0.016 Hz and 1000 Hz and sampled at 2500 Hz. We rejected electrodes whose impedance was more than 25 KΩ (4.0%, 3.4%, and 0.6% for healthy, MCI, and AD subjects, respectively). Impedance of the final set of electrodes was 5.5 ± 4.2, 5.9 ± 4.4, and 3.8 ± 3.5 KΩ for healthy, MCI, and AD subjects, respectively.

All subjects sat in a dark room in front of an LCD screen with their head supported by a chin rest. The screen (BenQ XL2411, resolution 1280 × 720 pixels, refresh rate 100 Hz) was gamma-corrected and placed at a mean ± SD distance of 58.1 ± 0.8 cm from the subjects (53.8–61.0 cm) according to their convenience (thus subtending a width of at least 52° and height of at least 30° of visual field for full-screen gratings). We calibrated the stimuli to the viewing distance in all cases.

Subjects performed a visual fixation task, as described in *Figure 1*. They performed the main 'Gamma' experiment in 2–3 blocks (total 543 blocks across 257 subjects) according to their comfort. We also tested 32 Hz SSVEPs on these subjects in the SSVEP experiment. We chose 32 Hz SSVEP (induced by gratings of temporal frequency of 16 cycles per second or cps) for two reasons. First, in a separate set of experiments, we had recorded spikes and local field potentials (LFP) in the primary visual cortex of awake monkeys while presenting counterphasing gratings at varying temporal frequencies and found that the SSVEP gain was highest for gratings with temporal frequencies of 12-16 cps (*Salelkar and Ray, 2020*). Second, 32 Hz was between slow and fast gamma bands, hence within the available time for the experiment, we were able to record SSVEP activity closest to both the gamma rhythms.

Subjects completed both experiments during a single session. We considered only those subjects for analysis in SSVEP experiment who had analyzable data for the Gamma experiment (see Artifact Rejection section below). This gave us a total of 222/11/5 subjects (99/3/2 females) for healthy/MCI/AD categories for the SSVEP experiment.

## Eye position analysis

We recorded eye signals (pupil position and diameter data) using EyeLink 1000 (SR Research Ltd) for all subjects (except for one subject each in healthy/MCI/AD categories). Eye data for Gamma

experiment is shown in *Figure 4*. We rejected stimulus repeats with fixation breaks (eye blinks or shifts in eye position outside a square window of width 5° centered on the fixation spot) during −0.5 s to 0.75 s of stimulus onset (mean ± SD: 16.7 ± 14.2%, 12.8 ± 12%, and 31.6 ± 18.4% for Gamma experiment; and 16.7 ± 15.1%, 9.1 ± 16%, and 46.4 ± 1% for SSVEP experiment, for healthy, MCI, and AD subjects, respectively). For the remaining repeats, all the subjects were able to maintain fixation with a standard deviation of less than 0.5°, 0.3°, and 0.6° for Gamma experiment and 0.6°, 0.4°, 0.6° for SSVEP experiment for healthy, MCI, and AD subjects, in either directions.

## Artifact rejection

We used a pipeline to reject artifact-containing data as described in *Murty et al., 2020*. Briefly, we applied a repeat-wise thresholding process on both time-domain waveforms and multitapered PSD (between −500 ms and 750 ms of stimulus onset) to select bad repeats across electrodes. We discarded those electrodes that had more than 30% of all repeats marked as bad, and subsequently labelled any repeat as bad if it occurred in more than 10% of total number of remaining electrodes. We next discarded those electrodes that had PSD slopes (calculated in 56-84 Hz range as described briefly in EEG data analysis subsection; see *Murty et al., 2020* for details) less than 0. Further, we discarded any block that did not have at least a single clean bipolar electrode pair (see Data Analysis subsection below) in any of the following three groups of bipolar electrodes: PO3-P1, PO3-P3, POz-PO3; PO4-P2, PO4-P4, POz-PO4 and Oz-POz, Oz-O1, Oz-O2. Despite these strict criteria, we ended up rejecting only 53/497, 4/31, and 1/15 blocks for healthy, MCI, and AD subjects; and we rejected only 5.5 ± 6.4%, 5.7 ± 3.6%, and 4.1 ± 1.9% of electrodes for healthy, MCI, and AD subjects, among those blocks that were analyzed. We then pooled data across all good blocks for each subject for final analysis. Those subjects who did not have any analyzable blocks (9 of 236 healthy subjects and 1 of 15 MCIs, respectively) were discarded from further analysis.

We used a similar procedure for the SSVEP experiment. Note that we did this experiment always towards the end, and therefore the signal quality could be poorer than the Gamma experiment. Consequently, we rejected 25/222, 5/12, and 3/5 blocks for healthy, MCI, and AD subjects; and 6.6 ± 7.2%, 8.5 ± 7.4%, and 5.7 ± 1% of electrodes among those blocks that were analyzed. Hence, we rejected data from 25/4/3 out of 222/11/5 subjects as they did not have any analyzable blocks, leaving 197/7/2 (93/1/1 females) healthy/MCI/AD subjects for analysis for the SSVEP experiment.

## EEG data analysis

For all analyses we re-referenced data at each electrode offline to its neighboring electrodes (bipolar reference). We thus obtained 112 bipolar pairs out of 64 unipolar electrodes (*Murty et al., 2020*). We considered the following nine bipolar electrodes for analysis: PO3-P1, PO3-P3, POz-PO3, PO4-P2, PO4-P4, POz-PO4, Oz-POz, Oz-O1, Oz-O2, which are inside the black encapsulation shown in *Figure 2D*. We discarded a bipolar electrode if either of its constituting unipolar electrodes was marked bad during artifact rejection. Data was pooled for the rest of the bipolar electrodes for further analysis.

We analyzed all data using custom codes written in MATLAB (The MathWorks, Inc, RRID:SCR_001622) as described in *Murty et al., 2020*. We computed PSD and the time-frequency power spectrograms using multi-taper method with a single taper using Chronux toolbox (*Mitra and Bokil, 2008*; http://chronux.org/, RRID:SCR_005547). We chose baseline period between −500 ms and 0 ms of stimulus onset, and stimulus period between 250 ms and 750 ms, to avoid stimulus onset-related transients, yielding a frequency resolution of 2 Hz for the PSDs. We calculated time frequency power spectra using a moving window of size 250 ms and step size of 25 ms, giving a frequency resolution of 4 Hz.

We calculated change in power in different frequency bands as follows:

$$\Delta Power = 10 \left( log_{10} \frac{\sum_f ST(f)}{\sum_f BL(f)} \right)$$

where *ST* and *BL* are stimulus and baseline power spectra (across frequencies of interest, *f*) averaged across all analyzable repeats and the nine bipolar electrodes as described above. As mentioned in

the Results, we computed the spectra for gratings with spatial frequency of 2 and 4 cpd and all four orientations for the Gamma experiment (unless otherwise mentioned). The total number of repeats that were thus analyzed were $184.2 \pm 58.3$, $174.2 \pm 52.1$, and $126.4 \pm 29.3$ for healthy, MCI, and AD subjects.

To test the dependence of alpha/gamma power on different orientations (see results), we calculated orientation selectivity for each subject using the following (*Murty et al., 2018*):

$$Orientation\,selectivity = \frac{|\sum\limits_{i=1}^{N} R_i e^{(j \cdot 2\theta_i)}|}{\sum\limits_{i=1}^{N} R_i}$$

where $\theta_i$ and $R_i$ are the orientations and absolute power ($\mu V^2$, for 250-750 ms after stimulus onset in the frequency band of interest), for each $i$ = [0° 45° 90° 135°] (thus, $N$ = 4). We averaged absolute power values across spatial frequencies for calculating the orientation selectivity.

For SSVEP experiment, we analyzed only the counter-phasing condition. There were $30.2 \pm 6.9$, $32.9 \pm 7.5$, and $17.5 \pm 2.1$ repeats for healthy, MCI, and AD subjects, respectively. We took the power at 32 Hz (twice the counter-phasing frequency, i.e., 16 cps) for analysis. Static gratings were presented in SSVEP experiment mainly to prevent adaptation and were discarded from analysis.

We generated scalp maps using the topoplot.m function of EEGLAB toolbox (*Delorme and Makeig, 2004*, RRID:SCR_007292), modified to show each electrode as a colored disc.

We calculated slopes (for *Figure 5—figure supplement 1*) by fitting baseline PSD averaged across all analyzable repeats and bipolar electrodes with a power law function using *fminsearch* in MATLAB: $P(f) = A.f^{-\beta}$, where $P$ is the PSD across frequencies $f$, $A$ is scaling factor, and $\beta$ is the slope.

## Microsaccades and pupil data analysis

We detected microsaccades using a threshold-based method described earlier (*Murty et al., 2018*), initially proposed by *Engbert, 2006*, for the analysis period of −0.5 s to 0.75 s of stimulus onset for the Gamma experiment. After removing the microsaccade-containing repeats ($85.6 \pm 47.6$, $76.5 \pm 36.5$, $70 \pm 17.2$), there were $98.5 \pm 46$ (minimum 7) repeats for healthy subjects (n = 226), $96.2 \pm 35.7$ (min: 54) for MCI (n = 11), and $64.3 \pm 14.4$ (min: 52) for AD subjects (n = 4), respectively, excluding three subjects for whom eye data could not be collected. We used coefficient of variation (CV, ratio of standard deviation to mean) of pupil diameter across time for every repeat as a measure of pupillary reactivity to stimulus for that repeat (*Murty et al., 2020*). We calculated CV for each analyzable repeat separately and calculated mean CV across repeats for every subject for comparison.

## Statistical analysis

All our statistical interpretations were based on non-parametric tests on medians using K-W test (results were similar if we used Wilcoxon sign-rank test). We used two-way ANOVA for comparing change in alpha/gamma power across spatial frequencies and orientations for 227 healthy subjects (see Results section). The study is controlled at $p<0.05$ regardless of the sample size, since the false alarm rate does not depend on the sample size.

## Bayes factor analysis

BF is the likelihood ratio of the marginal likelihood of alternate hypothesis to the likelihood of null hypothesis, given the data. This allows for comparing the alternate with null hypothesis, rather than just infer from the evidence for null hypothesis (*Jarosz and Wiley, 2014*; *Keysers et al., 2020*). In the present case, we used the Bayesian paired t-test with the Cauchy prior scaling set to 1. In this scheme, BF values below one suggests the absence of effect and evidence in favor of the null hypothesis, BF values between 1 and 3 provide anecdotal evidence in favor of the alternate hypothesis, while values between 3 and 10 provide substantial evidence and values above 10 provide strong evidence in favor of the alternative (*Jeffreys, 1998*). For gamma power, we calculated BF for right-tailed paired t-test (power(controls)>power(cases)), while for alpha power, we calculated BF for left-

tailed paired t-test (power(controls)<power(cases)). We used the MATLAB toolbox on BF by Bart Krekelberg (*Krekelberg, 2021*) based on *Rouder et al., 2012*.

## Regression analysis

We considered a linear regression model: $\Delta Power = \beta_0 + \beta_{CDR}.CDR + \beta_{AGE}.AGE + \beta_{GENDER}.GENDER + \varepsilon$. Categorical variable GENDER was converted to numerical variable by considering '0' for males and '1' for females. The variable AGE was considered as the subject's age in years, approximated to the integer. We used the *fitlm()* function in MATLAB to implement linear regression model. The function outputs significance (p-values) for the t-statistic of the null hypothesis test over coefficient of each corresponding regressor variable in the model.

## Data availability

All spectral analyses were performed using Chronux toolbox (version 2.10), available at http://chronux.org. Relevant data and codes are available at the following GitHub repository: https://github.com/supratimray/TLSAEEGProjectPrograms (*Murty, 2021* copy archived at swh:1:rev:5860e435fea06a49599ac81907bd63099e46581b).

## Acknowledgements

This work was supported by Tata Trusts Grant (to SR and NPR), Wellcome Trust/DBT India Alliance (intermediate fellowship 500145/Z/09/Z and senior fellowship IA/S/18/2/504003 to SR), and DBT-IISc Partnership Programme (to SR).

## Additional information

### Funding

| Funder | Grant reference number | Author |
|---|---|---|
| Tata Trusts | | Naren Prahalada Rao Supratim Ray |
| Wellcome Trust/DBT India Alliance | Intermediate fellowship 500145/Z/09/Z | Supratim Ray |
| Wellcome Trust/DBT India Alliance | Senior fellowship IA/S/18/2/504003 | Supratim Ray |
| DBT | | Supratim Ray |

The funders had no role in study design, data collection and interpretation, or the decision to submit the work for publication.

### Author contributions

Dinavahi VPS Murty, Conceptualization, Data curation, Formal analysis, Investigation, Visualization, Methodology, Writing - original draft, Writing - review and editing; Keerthana Manikandan, Data curation, Formal analysis, Investigation; Wupadrasta Santosh Kumar, Formal analysis, Writing - review and editing; Ranjini Garani Ramesh, Simran Purokayastha, Bhargavi Nagendra, Abhishek ML, Aditi Balakrishnan, Data curation, Investigation; Mahendra Javali, Validation, Writing - review and editing; Naren Prahalada Rao, Conceptualization, Resources, Data curation, Supervision, Funding acquisition, Validation, Methodology, Project administration, Writing - review and editing; Supratim Ray, Conceptualization, Resources, Data curation, Formal analysis, Supervision, Funding acquisition, Validation, Visualization, Methodology, Writing - original draft, Project administration, Writing - review and editing

### Author ORCIDs

Dinavahi VPS Murty  https://orcid.org/0000-0003-1726-5171
Keerthana Manikandan  https://orcid.org/0000-0002-6461-1930
Wupadrasta Santosh Kumar  https://orcid.org/0000-0001-7097-0414

Simran Purokayastha (ID) http://orcid.org/0000-0001-6096-1477
Supratim Ray (ID) https://orcid.org/0000-0002-1968-1382

### Ethics

Human subjects: We obtained informed consent from all participants before the experiment. The Institute Human Ethics Committees of Indian Institute of Science (IHEC numbers: original: 22/2014, revised: 7-15092017), NIMHANS, and M S Ramaiah Hospital, Bangalore approved all procedures.

### Decision letter and Author response

Decision letter https://doi.org/10.7554/eLife.61666.sa1
Author response https://doi.org/10.7554/eLife.61666.sa2

## Additional files

### Supplementary files

• Supplementary file 1. Supplementary tables. Supplementary table 1: Criteria used for consensus diagnosis of dementia. ACE: Addenbrooke's Cognitive Examination (*So et al., 2018*); CDR: Clinical Dementia Rating (*Hughes et al., 1982*; *Morris, 1993*); GPCOG: General Practitioner Assessment of Cognition (*Brodaty et al., 2002*); HAMD: Hamilton Depression Rating Scale (*Hamilton, 1960*; *Williams, 1988*); HMSE: Hindi Mental State Examination (*Ganguli et al., 1995*); IADL: Instrumental Activities of Daily Living (*Mathuranath et al., 2005*); NIA-AA: National Institute on Aging-Alzheimer's Association workgroups (*McKhann et al., 2011*); NPI: Neuropsychiatric Inventory (*Cummings et al., 1994*); TLSA: Tata Longitudinal Study of Aging. [1]NIA-AA criteria are presented from *McKhann et al., 2011*. [2]Cognitive decline could also be seen in delirium. So, this criterion is intended to rule out delirium. A combination of clinician assessment and HMSE is used. Acute change in cognitive status and/or HMSE <24 is suggestive of delirium. [3]This criterion is intended to rule out moderate/severe depression as that can cause cognitive impairment. Supplementary table 2. Criteria used for consensus diagnosis of probable AD. WMH: white matter hyperintensities; FTD: Frontotemporal dementia. The rest of the abbreviations are as described in Supplementary table 1. Supplementary table 3. Criteria used for consensus diagnosis of MCI. [1]Q4. Have there been some decline in memory in the past one year? (*Hughes et al., 1982*; *Morris, 1993*). Abbreviations are as described in Supplementary table 1.

• Transparent reporting form

• Reporting standard 1. STROBE statement.

### Data availability

All spectral analyses were performed using Chronux toolbox (version 2.10), available at http://chronux.org. Relevant data and codes are available at the following GitHub repository: https://github.com/supratimray/TLSAEEGProjectPrograms (copy archived at https://archive.softwareheritage.org/swh:1:rev:5860e435fea06a49599ac81907bd63099e46581b).

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

## Appendix 1

### Consensus diagnosis

As in *Table 1*, our major criteria to diagnose MCI/AD were clinical history and CDR, which were administered by a single clinician (neurologist or psychiatrist). However, to minimize observer bias, we followed the recommendations made in SAGES study (*Kimchi et al., 2017*) to arrive at a consensus diagnosis for each MCI/AD patient. We constituted a panel with four members, consisting of one neurologist (author MJ), two psychiatrists (authors AML and NPR), and one psychologist (author AB, an additional expert who did not participate in the single clinician diagnosis before). These members independently reviewed data variables available in TLSA (Tata Longitudinal Study of Aging) cohort to operationalize the NIA-AA criteria (*McKhann et al., 2011*) for probable AD and MCI. For each of the NIA-AA criteria, they operationalized an equivalent TLSA criterion (see Tables 1–3 in *Supplementary file 1*). Wherever longitudinal data was available, they considered trends over time for diagnosis.

They labelled a subject as MCI/AD only if all 4 of them concurred with the diagnosis. When they could not achieve at a consensus for any subject in the first instance (six subjects), they used Delphi method (*Dalkey and Helmer, 1963*; *Graham et al., 2003*) till all of them agreed upon the diagnosis. Briefly, the members were informed of the discrepancy within the panel, who then discussed among themselves and rated again. This process was iterated till all four members came to a consensus. Although they used this stringent approach to confirm the previous diagnoses, they reclassified as healthy only two subjects previously classified as MCI. They confirmed and retained initial diagnosis for rest of the 12/14 MCI and 6/6 AD subjects.

