## [Decision Letter]

**Acceptance summary:**

Murty and collaborators investigate alterations in stimulus-induced gamma (~30-80 Hz) oscillations in a geriatric population with early symptoms of cognitive decline and Alzheimer's disease (AD). Compared with sex and age-matched controls, the authors observed a significant and specific decrease in the stimulus-induced gamma-band frequency power in the occipital cortex. This suggests that reductions in stimulus-induced gamma oscillations are an electrophysiological marker of early stages of AD and cognitive decline. The relevance of these findings is further bolstered by previous studies in animal models showing cognitive improvement following optogenetic and flicker-induced increases of gamma power. The findings will impact Alzheimer's disease research in humans and in animal models.

**Decision letter after peer review:**

Thank you for submitting your article "Stimulus-induced gamma rhythms are weaker in human elderly with Mild Cognitive Impairment and Alzheimer's Disease" for consideration by *eLife*. Your article has been reviewed by 4 peer reviewers, one of whom is a member of our Board of Reviewing Editors, and the evaluation has been overseen by Laura Colgin as the Senior Editor. The following individual involved in review of your submission has agreed to reveal their identity: Conrado A Bosman (Reviewer #2).

The reviewers have discussed the reviews with one another and the Reviewing Editor has drafted this decision to help you prepare a revised submission.

Summary:

Murty and collaborators examined stimulus-induced gamma (~30-80 Hz) oscillations in a geriatric population with early symptoms of cognitive decline and Alzheimer's disease (AD). Compared with sex and age-matched controls, the authors observed a significant and specific decrease in the stimulus-induced gamma-band frequency power during the presentation of Cartesian grating stimuli. Conversely, they did not find a gamma power decrease when subjects performed a flickering experiment using static gratings presented at 32 Hz. Overall, this study addresses an important and timely question, because stimulus-induced gamma oscillations could be considered as an electrophysiological marker of early stages of AD, and previous studies in animal models have shown cognitive improvement following optogenetic and visually induced increases of gamma power.

All reviewers agreed that the study is interesting and addresses an important question. The main concerns focused on the sample size and stimulus variability. Please see below.

Essential revisions:

Statistics and study design

A main issue that was defined by the reviewers was the sample size used in this study. The authors should address several points:

1) The authors combine both groups to obtain significant effects which are not present in the MCI-group by itself. The text seems a bit fuzzy about this point, the statistics per group should be clarified or further analysed.

2) A problem is that clinical characteristics differ substantially between the two groups and the question remains whether abnormalities in gamma-band oscillations are present prior the onset of AD. This should be discussed/considered.

3) The authors chose appropriate non-parametric tests to perform statistical inference on groups with different samples. However, they should address the rather sever unbalanced effect sizes between the control and the condition group, providing, for example, some measurement of the distribution of gamma power between the two groups. The low number of cases with AD (5) is problematic because such a small sample is at the limit of the sensibility of a KW test.

4) A revised manuscript should explicitly acknowledge that, if it is impossible to increase the sample size due to COVID-19, the paper would benefit from including more subjects.

5) There are several instances where the authors included a subject (data) for one analysis and excluded the same subject for a different analysis. Specific examples; (1) The authors mention that two of the MCI patients (M4 and M8) were later re-evaluated and found to be cognitively normal. Removing their data for the individual-level comparison of gamma PSD did not cause the differences between the "cases" and "controls" to change much (Figure 3—figure supplement 1B), but it is unclear if their data were also removed from the group level data (i.e., Figure 2) and what happens to the conclusions made at the group level if their data are removed. (2) Grouped data in Figure 5A, 5C contains 12 cases but individual plots in Figure 5E contains only 11 cases ('A1 discarded'). We appreciate the authors providing the details about what is included/excluded in each analyses, but if a subject needed to be excluded because of some obvious reasons, then the authors should do it across all analysis to reduce the inconsistency in the data being analyzed.

6) It appears there is quite a clear increase in variability in the AD/MCI group as compared to the controls. Some subjects in fact show rather extremely strong gamma. It would be interesting to comment on this and quantify this.

7) Please provide dot-plots with the bar plots to see how the 6 AD subjects are scattered and the 14 MCI cases, and how robust the effects are.

8) Evaluation of diagnostic and inclusion criteria by an AD-expert would be recommended.

Stimuli

9) Is the difference between groups is driven by a specific spatial frequency and orientation of the gratings? During the experiment with gratings, the authors presented a sequence of gratings with different spatial frequencies. According to the manuscript, the presentation sequence and the number of trials with the same stimuli were fixed according to a preliminary experiment aiming to obtain the most responsive parameters per subject. However, it is well-known that stimulus properties affect the gamma power increase in visual areas. We suggest adding a control analysis comparing only those trials with the same type of gratings between subjects to discard that any differences due to stimuli properties are affecting these results.

10) The authors used multiple spatial frequencies and orientations for the 'gamma experiment', but they do not provide any frequency- or orientation-specific data, which makes it difficult to conclude if a specific frequency and/or orientation are driving the difference seen between the cases and the controls and thus more ideal than other combinations of frequencies and orientations. None of the spectrograms for 'gamma experiment' has any information about stimulus details. The authors are advised to provide more thorough quantitative analyses.

Data analysis

11) The authors do not analyze ERPs. Do those differ between subjects?

12) Does the difference between groups hold/changes if the gamma band is not broken up into slow and fast gamma band and instead analyzed as a single band (e.g., 30 – 80Hz).

13) The authors can maximize the utility of the microsaccades analysis used as a control. In the text, there is no context about why this analysis should be performed. Some authors have argued that a microsaccadic rhythm can modulate the strength of gamma-band power. It would be relevant to observe whether such microsaccades modulations can contribute differently to the strength of the gamma-band power across groups.

Questions about SSVEP experiments

14) What does it mean that there was a reduction in fast gamma (36-66Hz) when the real evoked signal was at 32Hz (and that 32Hz was not different between cases and controls) in SSVEP experiment?

15) There is a time lag between stimulus onset /off and spectral power change (>100ms) in both 'gamma experiment and SSVEP' but maybe more pronounced in SSVEP (Figure 5B). It would be informative if the authors can examine the spectral/waveform differences between control and cases within +/- 200 ms from stimulus onset (in other words, whether AD patients showed delayed response compared to controls?).

16) The authors should provide more explanation on how they extracted the "induced" gamma-band power from the SSVEP experiment (i.e., how they derived Figure 5—figure supplement 1A from Figure 5A). Were EEG signals phase-locked to the stimulation removed?

17) We presume the authors aimed to induce gamma oscillations at 32 Hz stimulation during the SSVP experiments. Yet, that number seems arbitrary. It may have been preferable to observe the same experiment across different flickering frequencies. It could help if the authors could show examples of ERP responses at the 32 Hz flickering, also analyzing the amplitude of the ERP components. A non-significant difference between amplitudes across groups would add evidence that the response observed is specific for gamma frequency-band and that it is not a consequence of this particular type of stimulation.

Discussion

18) The authors did not reference a study with seemingly contradictory findings (conducted by van Deursen et al.; J Neural Transm 2008), in which the gamma band (30 – 100 Hz) power induced by visual stimulation was actually higher in patients with AD at the occipital and parietal leads compared to the control subjects. The authors should discuss why their findings are different from those by van Deursen et al. and the potential implications of the differences.

19) Conceptually, the findings presented here may suggest that the gratings and SSVEP evoked gamma is sub-served by different mechanisms. The authors should discuss this.

[Editors' note: further revisions were suggested prior to acceptance, as described below.]

Thank you for submitting your article "Stimulus-induced gamma rhythms are weaker in human elderly with Mild Cognitive Impairment and Alzheimer's Disease" for consideration by *eLife*. Your article has been reviewed by 3 peer reviewers, and the evaluation has been overseen by Martin Vinck as the Reviewing Editor and Laura Colgin as the Senior Editor. The following individual involved in review of your submission has agreed to reveal their identity: Conrado A Bosman (Reviewer #2).

The reviewers have discussed their reviews with one another, and the Reviewing Editor has drafted this to help you prepare a revised submission. As you will see, most of the concerns have been addressed but a few concerns remain, as detailed below.

Essential revisions:

1) The relatively small sample size of the study remained a point of discussion among the reviewers, with consideration of the fact that the COVID situation prevented collecting a larger dataset. We have several recommendations related to the sample size:

i) It would be useful to point out the known fact that a small sample size does not increase the false alarm rate, hence the study is controlled at P<0.05 (regardless of the sample size).

ii) Given the small sample size, it would be good if the authors briefly justify the selection of the sample size. Was this a priori chosen, was there any power calculation?

iii) Reporting Bayes factor, esp. given that this is a medically relevant study, would be useful given the sample size.

2) The authors have adequately addressed major concerns in the revised manuscript. However, the authors need to address a few additional points. Especially, on Page 16, lines 348-352, and Page 17, lines 353-358, why do the authors think that their findings contradict findings of Adaikkan et al., Martorell et al., and Thomson et al.? The current findings are in no meaningful way comparable to Adaikkan et al., Martorell et al., and Thomson et al. Did the authors induce 40 Hz repeatedly and measured any cognitive and pathology outcomes? Or did Adaikkan et al., Martorell et al., showed SSVEP changes at 32 Hz in any human AD or mouse AD. The authors should provide a nuanced interpretation of their findings.

3) Page 5, lines 87-95, where is the corresponding graph or data points? If the authors would like to not include any additional graph, at least they should provide values for which they performed statistical analyses.

4) Given that the authors have CDR score for all subjects, it would be helpful to perform a regression analyses and provide an additional insight as to whether there is a correlation between CDR score and sensory evoked gamma. This is not an additional experiment but an additional analysis of the datasets to get an insight.

*Reviewer #2:*

The authors have addressed my comments satisfactorily and I endorse the publication of the manuscript as such. I congratulate the authors for their work.

*Reviewer #3:*

While I find the topic and approach taken in this study important as previously highlighted, I remain unconvinced that the current sample of MCI-participants (n = 12) and AD-patients (n = 5) is sufficiently large enough to allow for robust conclusions. It is unfortunate that in the current situation, additional data-collection is not immediately possible. However, this would be required to substantiate these findings. In the current format, the findings are promising pilot-data which require replication in a significantly larger sample. Accordingly, I cannot recommend publication of the article in *eLife*.

*Reviewer #4:*

The authors have adequately addressed major concerns in the revised manuscript. However, the authors need to address a few additional points. Especially, on Page 16, lines 348-352, and Page 17, lines 353-358, why do the authors think that their findings contradict findings of Adaikkan et al., Martorell et al., and Thomson et al.? The current findings are in no meaningful way comparable to Adaikkan et al., Martorell et al., and Thomson et al. Did the authors induce 40 Hz repeatedly and measured any cognitive and pathology outcomes? Or did Adaikkan et al., Martorell et al., showed SSVEP changes at 32 Hz in any human AD or mouse AD. The authors should provide a nuanced interpretation of their findings.

Other points:

1. Page 5, lines 87-95, where is the corresponding graph or data points? If the authors would like to not include any additional graph, at least they should provide values for which they performed statistical analyses.

2. Given that the authors have CDR score for all subjects, it would be helpful to perform a regression analyses and provide an additional insight as to whether there is a correlation between CDR score and sensory evoked gamma. This is not an additional experiment but an additional analysis of the datasets to get an insight.

---

## [Author Response]

Essential revisions:Statistics and study designA main issue that was defined by the reviewers was the sample size used in this study. The authors should address several points:1) The authors combine both groups to obtain significant effects which are not present in the MCI-group by itself. The text seems a bit fuzzy about this point, the statistics per group should be clarified or further analysed.

Thank you for raising this issue. We have now done the analysis on MCI and AD subjects separately (Figures 2 and 3); the analysis of MCI+AD data is only shown as a supplementary figure (Figure 3—figure supplement 3). Statistics have been clarified and presented uniformly.

2) A problem is that clinical characteristics differ substantially between the two groups and the question remains whether abnormalities in gamma-band oscillations are present prior the onset of AD. This should be discussed/considered.

Thank you for raising this issue. We now show reduction in gamma oscillations in MCI group separately, compared to their age and gender matched controls. We have presented these results in Figure 2 and have also included relevant statistics in the Results section.

3) The authors chose appropriate non-parametric tests to perform statistical inference on groups with different samples. However, they should address the rather sever unbalanced effect sizes between the control and the condition group, providing, for example, some measurement of the distribution of gamma power between the two groups. The low number of cases with AD (5) is problematic because such a small sample is at the limit of the sensibility of a KW test.

Thank you for raising this point. In our revised manuscript, we have now averaged the data from control subjects such that the final comparison is now done across equal samples (i.e. 12 MCIs and 12 averages of respective controls). Further, we have tested our results by limiting the number of control subjects that were averaged per case subject. The results continue to remain significant for several choices of this limit (we show results for N=1, and also present stats for N=4, which is the minimum number of controls for each case subject).

The low number of AD subjects is indeed problematic, but this number cannot be increased because of the pandemic. We note, however, that the results hold separately for MCIs, for which we have 12 subjects, and the results are robust and significant in the AD cohort (Figure 3) in spite of the small number of subjects.

4) A revised manuscript should explicitly acknowledge that, if it is impossible to increase the sample size due to COVID-19, the paper would benefit from including more subjects.

Thank you for this suggestion. We have acknowledged this limitation in the Results section.

5) There are several instances where the authors included a subject (data) for one analysis and excluded the same subject for a different analysis. Specific examples; (1) The authors mention that two of the MCI patients (M4 and M8) were later re-evaluated and found to be cognitively normal. Removing their data for the individual-level comparison of gamma PSD did not cause the differences between the "cases" and "controls" to change much (Figure 3—figure supplement 1B), but it is unclear if their data were also removed from the group level data (i.e., Figure 2) and what happens to the conclusions made at the group level if their data are removed. (2) Grouped data in Figure 5A, 5C contains 12 cases but individual plots in Figure 5E contains only 11 cases ('A1 discarded'). We appreciate the authors providing the details about what is included/excluded in each analyses, but if a subject needed to be excluded because of some obvious reasons, then the authors should do it across all analysis to reduce the inconsistency in the data being analyzed.

Thank you for raising this issue. We have followed your suggestion and have removed MCI subjects M4 and M8; and AD patient A1 (mentioned in the suggestion above) from all our analyses. Hence, now we have analyzed 12 MCIs and 5 ADs and have maintained consistency across the revised manuscript.

6) It appears there is quite a clear increase in variability in the AD/MCI group as compared to the controls. Some subjects in fact show rather extremely strong gamma. It would be interesting to comment on this and quantify this.

Thank you for raising this issue. The reason behind this apparent increase in variability is simply because each control data point is the average of many subjects. When we compare the variability of a single control subject versus case subject, the variability is actually comparable. We have now added this in the Results section.

7) Please provide dot-plots with the bar plots to see how the 6 AD subjects are scattered and the 14 MCI cases, and how robust the effects are.

Done.

8) Evaluation of diagnostic and inclusion criteria by an AD-expert would be recommended.

The expert panel, consisting of one neurologist (author MJ), two psychiatrists (authors AML and NPR) and one psychologist (author AB, an additional expert who did not participate in the single clinician diagnosis before), are all clinicians who have been involved in AD related research for several years.

Stimuli9) Is the difference between groups is driven by a specific spatial frequency and orientation of the gratings? During the experiment with gratings, the authors presented a sequence of gratings with different spatial frequencies. According to the manuscript, the presentation sequence and the number of trials with the same stimuli were fixed according to a preliminary experiment aiming to obtain the most responsive parameters per subject. However, it is well-known that stimulus properties affect the gamma power increase in visual areas. We suggest adding a control analysis comparing only those trials with the same type of gratings between subjects to discard that any differences due to stimuli properties are affecting these results.

Thank you for raising this point. Although gamma oscillations are indeed dependent on orientation, the orientation selectively is substantially weaker in EEG as compared to animal recordings (see Murty et al., 2018, JNS for a direct comparison between human and monkey data). We repeated this analysis here as well as found the same results: orientation only had a modest effect on gamma, and therefore we pooled the results across orientations. For spatial frequency, we did find that 2 and 4 CPD produced better results than 1 CPD. Therefore, we performed the main analyses using only 2 and 4 CPD. However, results remained qualitatively similar when 1 CPD was included. Spatial-frequency specific results have now been added as a separate supplementary figure (Figure 2—figure supplement 3 for MCI subjects and their controls; and Figure 3—figure supplement 2 for AD subjects and their controls).

We also note that for the main “Gamma” experiment, each subject (case and control) viewed all four orientations and three spatial frequencies, so the difference in gamma could not be due to any difference in stimuli between cases and controls. It is only for the SSVEP protocol that we fixed the SF and Ori to a single value in the interest of time.

10) The authors used multiple spatial frequencies and orientations for the 'gamma experiment', but they do not provide any frequency- or orientation-specific data, which makes it difficult to conclude if a specific frequency and/or orientation are driving the difference seen between the cases and the controls and thus more ideal than other combinations of frequencies and orientations. None of the spectrograms for 'gamma experiment' has any information about stimulus details. The authors are advised to provide more thorough quantitative analyses.

Thank you for this suggestion. As discussed in the previous point, gamma is not very orientation selective in EEG, and limiting ourselves to the two best spatial frequencies (2 and 4 CPDs) actually marginally improved the results. We have included this analysis in the Results section (first paragraph and Figure 2—figure supplement 3 and Figure 3—figure supplement 2).

Data analysis11) The authors do not analyze ERPs. Do those differ between subjects?

Thank you for this question. ERPs were not different between MCIs and controls. Specifically, we did not find any significant difference between P1/N1/P2 peak amplitudes. We have now added a figure showing these analyses (Figure 2—figure supplement 4).

12) Does the difference between groups hold/changes if the gamma band is not broken up into slow and fast gamma band and instead analyzed as a single band (e.g., 30 – 80Hz).

Thank you for this question. The results hold well for both 20-66 Hz and 30-80 Hz. We have added these in the Results section.

13) The authors can maximize the utility of the microsaccades analysis used as a control. In the text, there is no context about why this analysis should be performed. Some authors have argued that a microsaccadic rhythm can modulate the strength of gamma-band power. It would be relevant to observe whether such microsaccades modulations can contribute differently to the strength of the gamma-band power across groups.

Thank you for this suggestion. We have now added a supplementary figure to show the analysis of repeats that had micro saccades (Figure 4—figure supplement 2). We have also added the context and the appropriate reference (Yuval-Greenberg et al., 2008) in the Results section. Results remained similar with or without microsaccades containing trials.

Questions about SSVEP experiments14) What does it mean that there was a reduction in fast gamma (36-66Hz) when the real evoked signal was at 32Hz (and that 32Hz was not different between cases and controls) in SSVEP experiment?

Thank you for this question. For the SSVEP experiment, the real evoked signal was at 32 Hz. However, as noted in the reply to point 16 below, we did not calculate gamma power from the SSVEP experiment. Fast gamma power was calculated from the gamma experiment for the same set of subjects (7 MCIs) as for the SSVEP experiment, for comparing between the gamma and SSVEP results. Further, the differences in the trends of gamma and SSVEP suggest that these may have different origins and/or generating mechanisms in the brain. We have made these points clearer in the manuscript now.

15) There is a time lag between stimulus onset /off and spectral power change (>100ms) in both 'gamma experiment and SSVEP' but maybe more pronounced in SSVEP (Figure 5B). It would be informative if the authors can examine the spectral/waveform differences between control and cases within +/- 200 ms from stimulus onset (in other words, whether AD patients showed delayed response compared to controls?).

Thank you for this suggestion. SSVEP onset-related responses (0-250 ms) were comparable between MCIs and their respective controls as seen in the median time-courses of stimulus-induced change in power in 20-66 Hz frequency range. The RMS amplitude, peak amplitude and time of the peak of the onset response (calculated from trial-averaged time-courses for each subject) also did not differ among MCI subjects and their respective controls (KW test; χ^2^(13)=0.33/0.92/1.48, p=0.56/0.34/0.22 for RMS-amplitude/peak-amplitude/peak-time). We have now added a figure (Figure 6—figure supplement 2D-E) and the respective analyses in the Results section.

16) The authors should provide more explanation on how they extracted the "induced" gamma-band power from the SSVEP experiment (i.e., how they derived Figure 5—figure supplement 1A from Figure 5A). Were EEG signals phase-locked to the stimulation removed?

Thank you for pointing this out. We have not extracted induced gamma-band power from the SSVEP experiment. Instead, we had repeated the analysis done in Figure 2 for those subjects who had undergone the SSVEP experiment, for a comparison. We have made this clearer in the manuscript.

17) We presume the authors aimed to induce gamma oscillations at 32 Hz stimulation during the SSVP experiments. Yet, that number seems arbitrary. It may have been preferable to observe the same experiment across different flickering frequencies.

Thank you for raising this issue. We could not include more counter-phasing gratings for the study due to time constraints involved in the project. As we were working with elderly population in collaboration with clinical teams, we had strict timelines. However, we chose 32 Hz SSVEP (induced by gratings of temporal frequency of 16 cycles per second) for two reasons. First, in a different study, we recorded spikes and local field potentials (LFP) in the primary visual cortex of awake monkeys. We found that the SSVEP gain was highest for gratings with temporal frequencies of 12 to 16 cps (Salelkar and Ray, 2020). Second, 32 Hz was between the slow and fast gamma bands, hence within the available time, we were able to record SSVEP activity closest to both the gamma rhythms. We have now added these points in the Methods section.

It could help if the authors could show examples of ERP responses at the 32 Hz flickering, also analyzing the amplitude of the ERP components. A non-significant difference between amplitudes across groups would add evidence that the response observed is specific for gamma frequency-band and that it is not a consequence of this particular type of stimulation.

Thank you for this suggestion. We performed time-frequency analysis on trial-averaged time-amplitude waveforms for each subject instead for averaging time-frequency spectra of individual repeats for each subject, as done for SSVEP analysis in Figure 6. Trends were not different between either analyses. We have now summarized the results in Figure 6—figure supplement 2.

Discussion18) The authors did not reference a study with seemingly contradictory findings (conducted by van Deursen et al.; J Neural Transm 2008), in which the gamma band (30 – 100 Hz) power induced by visual stimulation was actually higher in patients with AD at the occipital and parietal leads compared to the control subjects. The authors should discuss why their findings are different from those by van Deursen et al. and the potential implications of the differences.

Thank you for this suggestion. Our findings are different from the study of van Deursen et al., probably due to the differences in task settings: we had tested for visual gamma rhythms induced by gratings, whereas they had tested for resting-state gamma as well as gamma activity for multiple colored objects randomly moving on a computer screen (taken from screensavers). Whether our results hold true for drifting (moving) and chromatic gratings needs to be tested in future studies. We have added these points and the reference in the Discussion section now.

19) Conceptually, the findings presented here may suggest that the gratings and SSVEP evoked gamma is sub-served by different mechanisms. The authors should discuss this.

Thank you for this suggestion. We have added this in the Discussion section now. However, we have avoided extensive discussion because the comparison between SSVEP and gamma is based on limited data and therefore we did not want to speculate too much.

[Editors' note: further revisions were suggested prior to acceptance, as described below.]

Essential revisions:1) The relatively small sample size of the study remained a point of discussion among the reviewers, with consideration of the fact that the COVID situation prevented collecting a larger dataset. We have several recommendations related to the sample size:i) It would be useful to point out the known fact that a small sample size does not increase the false alarm rate, hence the study is controlled at P<0.05 (regardless of the sample size).

Thank you for this suggestion. We have included this statement in Statistical analysis section of Methods.

ii) Given the small sample size, it would be good if the authors briefly justify the selection of the sample size. Was this a priori chosen, was there any power calculation?

Thank you for pointing this out. This was a community based study, in which subjects were recruited from the community based on advertisements, without any prior knowledge of their cognitive status. Their clinical identity was determined by trained clinicians and psychologists through various tests (as described in the paper) and was revealed to us after the experiments were over. Therefore, no power calculation was done. We have made this clear in the Methods section.

iii) Reporting Bayes factor, esp. given that this is a medically relevant study, would be useful given the sample size.

Thank you very much for this excellent suggestion. We performed this analysis and found that the results were highly consistent with our previous statistical tests. In summary, for the MCI group (N=12), the Bayes factor (BF) was ~1.89 for both slow and fast gamma when standard frequency ranges were used. However, as noted in the paper, we used fixed frequency ranges for the two gamma bands even though the difference between cases and controls was visible only in the middle of the bands. Simply choosing more “sensitive” gamma ranges (26-34 Hz for slow gamma and 44-56 Hz for fast gamma) yielded BFs of 3.34 and 4.6, thereby providing substantial evidence in favour of the alternate hypothesis even when only MCIs were considered. Upon considering both MCI and AD groups together (N=17), the BFs were 13.70 and 8.60 for the two gamma bands, providing strong evidence in favour of the alternate over the null hypothesis.

2) The authors have adequately addressed major concerns in the revised manuscript. However, the authors need to address a few additional points. Especially, on Page 16, lines 348-352, and Page 17, lines 353-358, why do the authors think that their findings contradict findings of Adaikkan et al., Martorell et al., and Thomson et al.? The current findings are in no meaningful way comparable to Adaikkan et al., Martorell et al., and Thomson et al. Did the authors induce 40 Hz repeatedly and measured any cognitive and pathology outcomes? Or did Adaikkan et al., Martorell et al., showed SSVEP changes at 32 Hz in any human AD or mouse AD. The authors should provide a nuanced interpretation of their findings.

Thank you for raising this point. As the reviewer has rightly pointed out, we cannot draw a direct comparison of our study with the studies cited above. This is because the model organism as well as the entrainment frequencies are different in our study. We have now made this point clear in the Discussion section.

3) Page 5, lines 87-95, where is the corresponding graph or data points? If the authors would like to not include any additional graph, at least they should provide values for which they performed statistical analyses.

Thank you for pointing this out. We have now made a separate figure (Figure 1—figure supplement 1) showing these data.

4) Given that the authors have CDR score for all subjects, it would be helpful to perform a regression analyses and provide an additional insight as to whether there is a correlation between CDR score and sensory evoked gamma. This is not an additional experiment but an additional analysis of the datasets to get an insight.

Thank you for this suggestion. We have included this regression analysis, and it does show a significant effect for both gamma ranges (slow gamma: β_CDR_ = -0.57, p = 0.0017; fast gamma: β_CDR_ = -0.25, p = 0.0063) but not for the alpha range, consistent with other results presented earlier.